medicinal chemistry/synthetic chemistry

glucopyanosyl-conjugated benzyl, colon cancer, HCT-116, 293T

**Authors for correspondence:**
Boqiao Fu
e-mail: fuboqiao@126.com
Caiqin Qin
e-mail: qincq@hbeu.edu.cn

[†]These authors contributed equally to this work.
[‡]Present address: College of Chemistry and Material Sciences, Hubei Engineering University, Xiaogan 432000, People's Republic of China.

This article has been edited by the Royal Society of Chemistry, including the commissioning, peer review process and editorial aspects up to the point of acceptance.

# Synthesis and pharmacological characterization of glucopyranosyl-conjugated benzyl derivatives as novel selective cytotoxic agents against colon cancer

Boqiao Fu[1,†,‡], Yingjie Li[2,†], Shaoyong Peng[2], Xiaolin Wang[2], Jingying Hu[3], Long Lv[1], Caifen Xia[1], Dai Lu[4] and Caiqin Qin[1]

[1]Hubei Provincial Collaborative Innovation Center of Biomass Resources Transformation and Utilization, College of Chemistry and Materials Science, Hubei Engineering University, Hubei 432000, People's Republic of China
[2]Guangdong Institute of Gastroenterology, Guangdong Provincial Key Laboratory of Colorectal and Pelvic Floor Diseases, The Sixth Affiliated Hospital, Sun Yat-sen University, Guangzhou, Guangdong 510655, People's Republic of China
[3]State Key Laboratory of Oncogenes and Related Genes, Shanghai Cancer Institute, Renji Hospital, Shanghai Jiaotong University School of Medicine, Shanghai 200032, People's Republic of China
[4]Department of Pharmaceutical Sciences, Rangel College of Pharmacy, Texas A&M University, TX 78363, USA

BF, 0000-0002-7022-6051

Glucopyranosyl-conjugated benzyl derivatives containing a [1,2,3]-triazole linker were synthesized. Benzyl served as an important pharmacophore in anti-cancer compounds. Compound **8d** inhibited the proliferation of colorectal cancer cells with the potency comparable to 5-fluorouracil (5-FU) with improved selectivity towards cancer cells. The antiproliferative activity of **8d** is achieved through triggering apoptotic cell death.

## 1. Introduction

Colorectal cancer (CRC) is the third most common type of cancer. 5-Fluorouracil (5-FU) is the most common chemotherapeutic drug

**Scheme 1.** Some representative compounds with benzyl group (in blue) as pharmacophore.

for CRC, which is a thymidylate synthase inhibitor limiting the rate of pyrimidine nucleotide synthesis. Treatment with 5-FU prolongs the median survival of advanced CRC patients from approximately 6 months to about 11 months [1]. However, 5-FU showed poor selectivity towards normal cells, leading to certain common toxicities, such as neutropenia, stomatitis, diarrhoea or palmar–plantar erythrodysesthesia [1]. Developing more effective and selective chemotherapeutics for CRC treatment is highly relevant to the improvement of the survival and the quality of life of CRC patients. Benzyl and its derivatives are important pharmacophore in many anti-cancer compounds (scheme 1) [2–5]. To investigate whether the cytotoxicity of benzyl can be used for therapeutic gain, we designed and synthesized carbohydrate conjugates of benzyl (**8a–8i**) to study their anti-cancer effects in CRC cells.

Carbohydrates are the most abundant biomolecules, which play critical roles in many cellular responses such as cell recognition, signalling to other cellular molecules and cell adhesion [6]. Compared with normal cells, cancer cells typically have increased glucose transporters (GLUTs) on their membrane surface to meet the increased energy demand in proliferation. This leads to carbohydrates uptake significantly higher than normal cells. Hence, it renders the carbohydrate-conjugated cytotoxic agents selectivity towards cancerous cell over normal cells [7]. Taking advantage of this phenomenon, a number of chemotherapeutics have been conjugated with monosaccharide, such as doxorubicin, daunorubicin, epirubicin, pentostatin, etc. [6]. Additionally, carbohydrate moiety can increase the solubility of the resultant conjugates with cytotoxic agents. The conjugation of carbohydrates with pharmacologically important compounds to form glycoconjugates is an important approach for drug discovery and development [1,8–10]. The conjugation of carbohydrates with cytotoxic agents is frequently achieved through click chemistry, which typically involved the click reaction between an alkynyl-substitited cytotoxic warhead with an azido-containing carbohydrate [11]. Although a [1,2,3]-triazole linker is formed during this coupling, the [1,2,3]-triazole moiety generally does not reduce the cytotoxicity, which has been demonstrated in numerous cases. For instance, cyclopentylamine with different structural units of [1,2,3]-triazole exhibit improved antiproliferation against lung cancer cells than cyclopentylamine itself [12]. A group of 4β-[(4-alkyl)-1,2,3-triazol-1-yl] podophyllotoxin derivatives with various alkyl or hydroxyl groups on the triazole moiety robustly enhanced the cytotoxicity against several types of cancer cells [13]. Furthermore, the

cytotoxic novobiocin analogues synthesized by 'click chemistry' exhibited potent cytotoxicity for breast cancer cells through downregulating heat shock protein 90 [14]. The [1,2,3]-triazole tethered imidazole–isatin conjugate showed that introducing the [1,2,3]-triazole linker did not abolish the cytotoxicity against breast cancer cells [15]. In the modification of lavendustin C, which is a plant-derived anti-cancer agent, replacing its methylene amino moiety (CH$_2$–NH) with [1,2,3]-triazole ring maintained the anti-cancer effects [16]. Collectively, these results indicated that click chemistry product from coupling a carbohydrate with a cytotoxic warhead is a viable approach for developing novel anti-cancer agents with the promise to improve drugs safety and bioavailability. Using this strategy, we previously coupled cytotoxic agent *p*-aminophenyl arsenoxide (*p*-APAO) with a glucopyranose through a [1,2,3]-triazole linker [17]. It was found that the conjugate significantly increased the antiproliferative selectivity between CRC cells and normal cells [17]. In this study, we hypothesized that conjugating the cytotoxic benzyl with a carbohydrate could yield novel cytotoxic compounds with improved cytotoxic selectivity towards cancer cells. A previous study demonstrated that 1,3,4,6-tetra-*O*-acetyl-D-glucose-conjugated indoloquinoline derivatives exhibited more optimal anti-cancer activities than their deacetylation counterparts [18]. Similarly, our early work in conjugation of 1,3,4,6-tetra-*O*-acetyl-D-glucose with organic arsenic compound (*p*-APAO) suggested that the presence of four acetyl groups on the β-D-glucose still maintained the cytotoxic activity of the organic arsenic compound [17]. Therefore, we set forth to explore the modification at 2-position carbon of 1,3,4,6-tetra-*O*-acetyl-D-glucose with benzyl to investigate the anti-cancer potential of carbohydrate-conjugated benzyl.

# 2. Results and discussion

## 2.1. Synthesis of glucopyranosyl-conjugated benzyl derivatives

In this study, the designed compounds **8a–i** were synthesized through the click reaction (scheme 2). The key intermediate 1,3,4,6-tetra-*O*-acetyl-2-azido-2-deoxy-D-glucose **4** was prepared from commercially available β-D-glucosamine hydrochoride **1**, of which the 2-amino group was converted into azido group via reaction with imidazole-1-sulfonyl azide hydrochloride **2** under basic condition [19,20]. The propargyl-functionalized benzyl **7** was prepared via alkylation of the benzyl alcohol **6** with 3-bromopropyne [18–31]. Coupling of compound **4** with the propargyl-functionalized benzyl **7** in the presence of copper(I) at 100°C with the assistance of microwave irradiation yielded the desired compounds **8a–i**. All the products were purified by silica gel chromatography and charaterized by [1]H NMR, [13]C NMR and HRMS. In order to research the effect of the acetyl groups of compound **8d**, compound **9d** was prepared by deacetylation of compound **8d** under the solution of the mixture of methanol, water and triethylamine [32]. The synthetic chemistry employed in this work is illustrated in scheme 2.

## 2.2. The cytotoxicity of the compounds against HCT-116 and 293T

The synthesized compounds were evaluated for their antiproliferation activity against CRC cell line HCT-116 via a MTS assay. The human embryonic kidney 293T cells were used as a model of normal cells to assess the selectivity between normal cells and cancer cells. The cytotoxicity results shown in table 1 indicated that the benzyl alcohol groups such as **6b** and **6d** and peracetyl glucose like **4** and 2-amino-1,3,4,6-*O*-acetyl-D-glucose (AOAG) had almost no cytotoxicty towards HCT-116 and 293T. The antiproliferation activity was fairly weak in the presence of the triazole group (i.e. **8a** versus **8b**). In addition to this result, the data from compounds **8b–8i** indicated that the benzyl group plays a pivotal role to induce the cytotoxic activity of this series of compounds. Interestingly, the para substitution significantly impacts the cytotoxicity towards normal cells. In the absence of a substitent at the *para*-position of the phenyl ring, the compound is farily toxic to normal cells (e.g. **8b** versus **8c–i**). The compounds with electron-donating groups such as the methyl group (**8c**) and OMe (**8d**) showed enhanced selectivity between the normal cells and the cancer cells. Stronger electron-donating effect leads to better selectivity (**8d** versus **8c**). When electron-withdrawing groups were introduced at the *para*-position, the cytotoxicity selectivity between normal cells and cancer cells is greatly reduced. It showed a 23-fold difference (e.g. **8d** versus **8i**). In comparison with the non-substituted analogue **8b**, compound **8d** exhibited 64-fold enhancement of cytotoxic selectivity towards cancer cells.

**Scheme 2.** Synthesis of glucopyranosyl-conjugated benzyl derivatives. Reaction conditions: (a) 2, $K_2CO_3$, MeOH, rt; (b) pyridine, $Ac_2O$; (c) NaH, THF; (d) 4, sodium ascorbate, copper sulfate pentahydrate, microwave, 100°C, N, N-dimethylforamide; (e) MeOH, water and $NEt_3$ (the ratio of volume is 5 : 4 : 1), reflux, 4 h.

**Table 1.** The cytotoxicity of the derivatives on HCT-116 and 293T cells.

| | IC$_{50}$ (µM) | | |
| --- | --- | --- | --- |
| | HCT-116 | 293T | selectivity[a] |
| 5-FU | 3.62 ± 1.22 | 9.13 ± 1.63 | 2.52 |
| AOAG[b] | >200 | >200 | ND[c] |
| **4** | 97.48 ± 1.58 | 46.90 ± 2.93 | 0.48 |
| **6b** | >200 | >200 | ND[c] |
| **6d** | >200 | >200 | ND[c] |
| **8a** | >50 | >50 | ND[c] |
| **8b** | 4.23 ± 1.98 | 0.74 ± 0.45 | 0.17 |
| **8c** | 6.40 ± 0.20 | 19.57 ± 0.13 | 3.06 |
| **8d** | 2.88 ± 0.32 | 31.35 ± 0.10 | 10.89 |
| **8e** | 4.17 ± 0.06 | 8.10 ± 0.08 | 1.94 |
| **8f** | 7.04 ± 0.06 | 22.41 ± 0.05 | 3.18 |
| **8 g** | 4.55 ± 0.23 | 11.67 ± 0.16 | 2.56 |
| **8 h** | 14.33 ± 2.18 | 6.09 ± 0.42 | 0.42 |
| **8i** | 17.29 ± 3.86 | 8.00 ± 0.77 | 0.46 |
| **9d** | >200 | >200 | ND[c] |

[a]Selectivity: the ratio of cytotxicity against 293T cells over the cytotoxicity against HCT-116 cells.
[b]2-amino-1,3,4,6-O-acetyl-D-glucose.
[c]ND, not determined.

The results of compound **9d** indicated that acetyl group played an important role in the anti-cancer effects of compound **8d**. Deacetylate significantly reduced the anti-tumour activity as observed from compound **9d**. Acetyl was helpful to increase the anti-tumour activity via improving liposolubility [18].

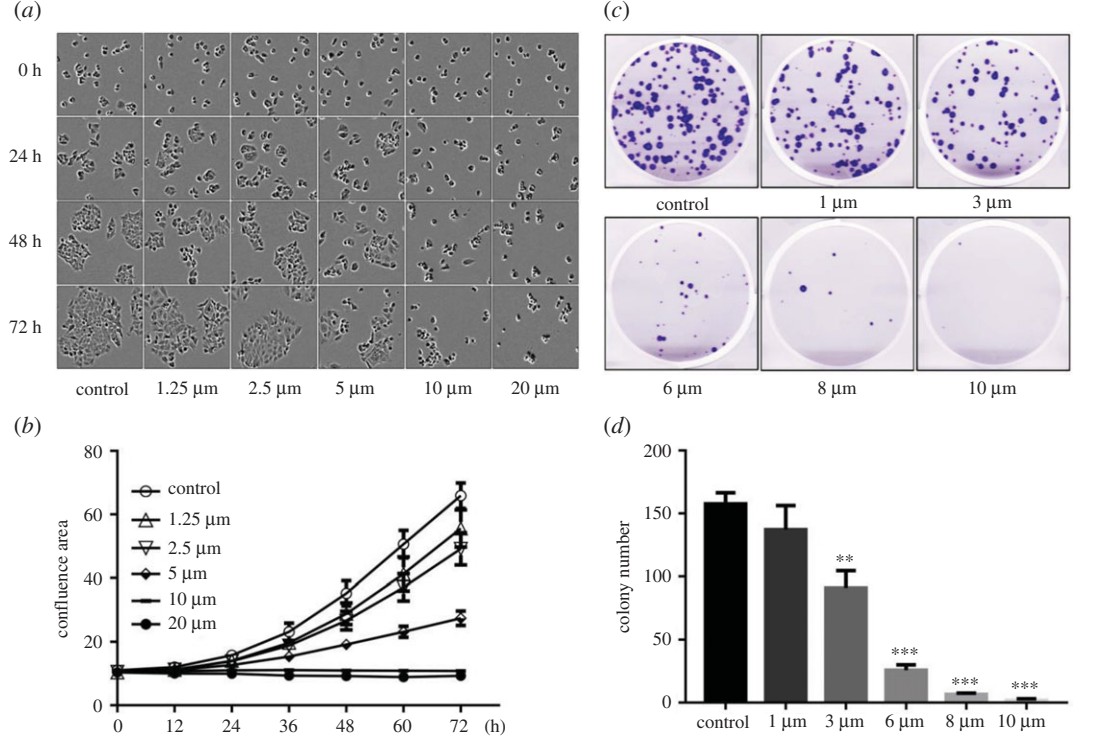

**Figure 1.** Compound **8d** inhibits the proliferation of HCT-116 cells. (*a*) Representative sections of compound **8d**-treated HCT-116 cells showing confluence area detected by Incucyte system. (*b*) Quantitative data of HCT-116 cell confluence by confluence area calculated module in Incucyte system. (*c*) Representative wells of HCT-116 clones treated with or without compound **8d**. (*d*) Colony number was quantified in the imaging sections by ImageJ. (*$p < 0.05$, **$p < 0.001$ and ***$p < 0.0001$ versus control.)

## 2.3. Compound **8d** inhibits the proliferation of HCT-116 cells

Compound **8d** exhibited an antiproliferative activity that is comparable to 5-FU against CRC cells HCT-116 and the best selectivity index within this series of compounds. Thus, it was further studied to understand its underlying antiproliferation mechanism. The confluence areas of HCT-116 cells treated with various concentrations of compound **8d** were detected by a live cell analysis imaging system. As shown in figure 1*a,c,d*, compound **8d** dose-dependently inhibited the proliferation of cells at various concentrations. The results showed that the proliferation of the HCT-116 cells was completely inhibited by 10 μM of **8d** for at least 72 h (figure 1*b*). Compound **8d** also inhibited the colony formation of HCT-116 by a dose-dependent manner after pre-treatment of cells with compound **8d** in various concentrations for 24 h (figure 1*c,d*). These results indicated that compound **8d** has a potent inhibitory effect on HCT-116 cell growth.

## 2.4. Compound **8d** triggers apoptotic cell death in HCT-116

Next, the type of cell death induced by compound **8d** was investigated. Apoptotic DNA fragmentation in compound **8d**-treated cells was detected by Hoechst 33258 staining (indicated by arrows, figure 2*a*). Apoptotic cell death was further confirmed by TUNEL assay that DNA fragmentation emerged in compound **8d**-treated cells (as indicated by arrows, figure 2*b*). Annexin V staining in living cells showed that Annexin-V-positive staining emerged on cellular membranes in agent-treated cells. Annexin V/PI staining assay were carried out to analyse the apoptotic rate of HCT-116 cells. As shown in figure 2*c*, compound **8d** dose-dependently induced early apoptosis and late apoptosis at concentrations of 2.5 and 5 μM. The results indicated that compound **8d** inhibits cell proliferation by inducing apoptotic cell death rather than necrosis.

## 3. Conclusion

A series of glucopyranosyl-conjugated benzyl derivatives were designed and synthesized, one of which showed antiproliferative activity comparable to 5-FU against CRC cells HCT-116 with improved selectivity

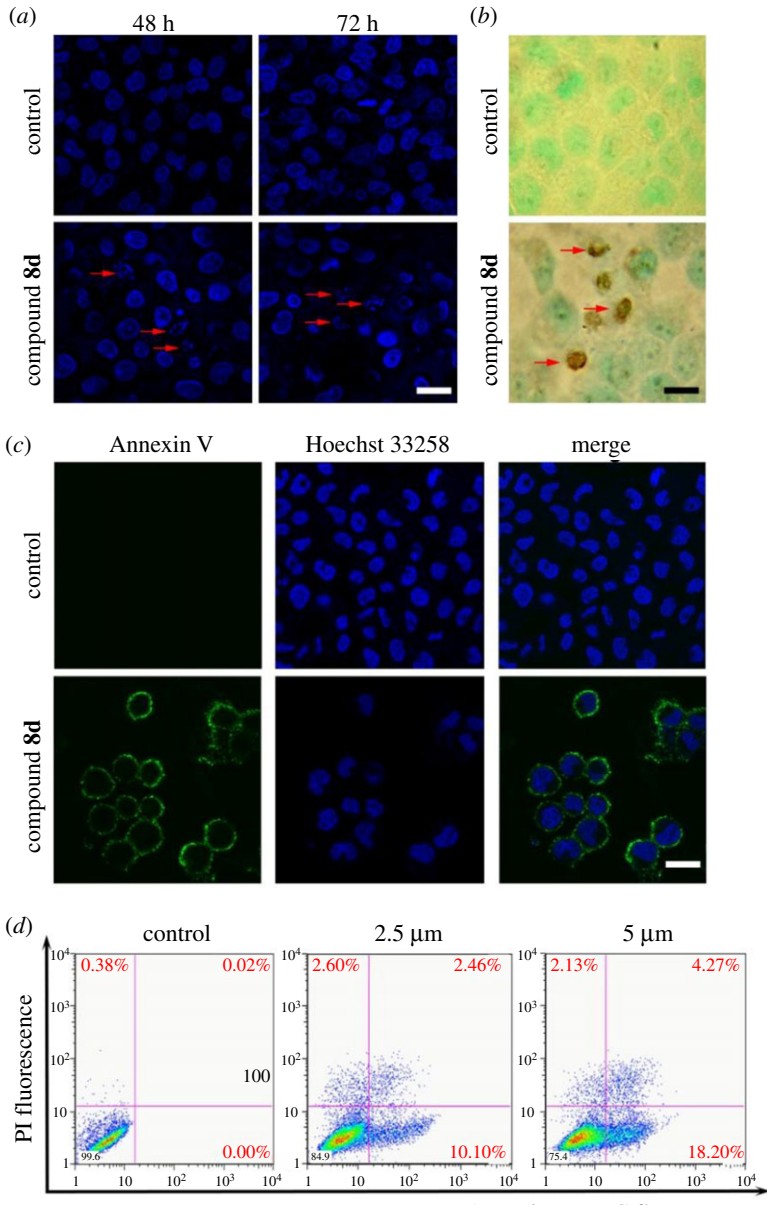

**Figure 2.** Compound **8d** induces apoptotic cell death in HCT-116 cells. (*a*) Representative nuclear morphometry stained with Hoechst 33258 in HCT-116 cells treated with or without compound **8d** for 48 and 72 h; the scale bar represents 10 μm, ×400 magnification. (*b*) Representative TUNEL staining in HCT-116 cells treated with or without compound **8d** for 72 h; the scale bar represents 10 μm, ×400 magnification. (*c*) Representative Annexin V-FITC staining in HCT-116 cells treated with compound **8d** for 72 h; the scale bar represents 10 μm, ×400 magnification. (*d*) The rate of apoptotic death cells after compound **8d** treatment was quantified by the Annexin V/PI assay.

over normal cells modelled with 293T cells. The substitution at the *para*-position of the benzyl moiety significantly influenced the therapeutic index of the benzyl-carbohydrate conjugate. The anti-cancer effects of this compound (**8d**) were shown to induce apopototic cell death. The antiprolifertive effect is long lasting in the *in vitro* study. Further investigation of its novel anti-cancer agent is warranted in animal models of CRCs.

# 4. Experimental section

## 4.1. General chemistry

All the chemical reagents and solvents were purchased from Sinopharm Group Company limited, and used without further purification, unless specified otherwise. Propargyl alcohol was purchased from

Xiya Reagent (Shangdong, China). All anhydrous reactions were performed under nitrogen atmosphere. Organic phases during work-up were dried over anhydrous $Na_2SO_4$ and solvents were removed by evaporation under reduced pressure.

The intermediates and products were purified by preparation thin-layer chromatography (TLC) or column chromatography. TLC was carried out by silica gel F254, and column chromatography was conducted over silica gel (200–300 mesh), both of which were obtained from Qingdao Ocean Chemicals (Qingdao, China). In all experiments, water used was distilled and purified by a Milli-Q system (Millipore, USA). [1]H NMR and [13]C NMR spectra of final compounds were recorded on a Bruker Ultrashield 400 MHz Plus spectrometer using TMS as an internal standard. All chemical shifts are reported in the standard $\delta$ notation of parts per million. High-resolution mass spectra were analysed using Waters UPLC Class I/XevoG2Q-Tof. Microwave reactions were performed using AntonPaar Monowave 300 microwave reactor (Austria).

### 4.1.1. 1,3,4,6-Tetra-O-acetyl-2-azido-2-deoxy-D-glucose (4)

Compound **4** was synthesized in yield of 70% ($\beta : \alpha = 3:1$) and the intermediate **3** without further purification was used for next step directly according to the literature [19–21]. The NMR and mass data of **6** were in agreement with the data reported in the literature [19–21].

### 4.1.2. General procedure A: preparation of compounds 7b–7i

Compound **6** (2.17 mmol) was dissolved in 10 ml dried tetrahydrofuran under $N_2$ and ice-bath for 30 min. Then 60% sodium hydride (104 mg, 2.605 mmol) was added to the solution under ice-bath. The solution was stirred for 30 min. 3-Bromopropyne (0.28 ml, 3.257 mmol) was added into the reaction mixture solution cooled with ice-bath. The reaction was monitored by TLC. Upon completion of the reaction, the reaction was quenched with water and the resultant mixture was extracted with ethyl acetate. The organic layers were combined and dried over anhydrous sodium sulfate. Filtration and removal of solvent *in vacuo* provided the crude product, which was purified through silica gel chromatography.

#### 4.1.2.1. ((Prop-2-yn-1-yloxy)methyl)benzene (7b)

Compound **7b** was synthesized according to the General Procedure A in the yield of 64.3%. [1]H NMR (CDCl₃, 400 MHz) $\delta$: 7.35(m, 5H, Ph-H), 4.60 (s, 2H, O-CH₂-Ph), 4.17 (s, 2H, O-CH₂-alkyne), 2.47 (s, 1H, H of terminal alkyne). The data of **7b** were in agreement with the data reported in the literature [22].

#### 4.1.2.2. 1-Methyl-4-((prop-2-yn-1-yloxy)methyl)benzene (7c)

Compound **7c** was synthesized according to the General Procedure A in the yield of 80%. [1]H NMR (CDCl₃, 400 MHz) $\delta$: 7.24 (d, $J$ = 7.6 Hz, 2H, Ph-H), 7.16 (d, $J$ = 7.6 Hz, 2H, Ph-H), 4.57 (s, 2H, O-CH₂-Ph), 4.14 (d, $J$ = 2.4 Hz, 2H, O-CH₂-alkyne), 2.45 (s, 1H, H of terminal alkyne), 2.34 (s, 3H, CH₃); [13]C NMR (CDCl₃, 100 MHz) $\delta$: 137.7, 134.2, 129.2, 128.3, 71.4, 56.8, 21.2; HRMS (ESI). M + H⁺: $C_{11}H_{13}O$, calcd for: 161.0966; found: 161.0966. The data of **7c** were in agreement with the data reported in the literature [23].

#### 4.1.2.3. 1-Methoxy-4-((prop-2-yn-1-yloxy)methyl)benzene (7d)

Compound **7d** was synthesized according to the General Procedure A in the yield of 80%. [1]H NMR (CDCl₃, 400 MHz) $\delta$: 7.30 (d, $J$ = 8.46 Hz, 2H, Ph-H), 6.89 (d, $J$ = 8.46 Hz, 2H, Ph-H), 4.56 (s, 2H, O-CH₂-Ph), 4.15 (m, 2H, O-CH₂-alkyne), 3.82 (s, 3H, OCH₃), 2.47 (m, 1H, H of terminal alkyne); [13]C NMR (CDCl₃, 100 MHz) $\delta$: 159.4, 129.9, 129.3, 113.9, 71.1, 56.7, 55.3; HRMS (ESI). M + H⁺: $C_{11}H_{13}O_2$, calcd for: 177.0916; found: 177.0916. The data of **7d** were in agreement with the data reported in the literature [24,25].

#### 4.1.2.4. 1-Fluoro-4-((prop-2-yn-1-yloxy)methyl)benzene (7e)

Compound **7e** was synthesized according to the General Procedure A in the yield of 80%. [1]H NMR (CDCl₃, 400 MHz) $\delta$: 7.36–7.33 (m, 2H, Ph-H), 7.07–7.03 (m, 2H, Ph-H), 4.59 (s, 2H, O-CH₂-Ph), 4.18 (m, 2H, O-CH₂-alkyne), 2.49 (m, 1H, H of terminal alkyne); [13]C NMR (CDCl₃, 100 MHz) $\delta$: 163.7,

161.3, 133.0, 130.0, 115.5, 115.3, 70.8, 57.1. HRMS (ESI). M+H$^+$: $C_{10}H_{10}FO$, calcd for: 165.0716; found: 165.0713. The data of **7e** were in agreement with the data reported in the literature [26].

### 4.1.2.5. 1-Chloro-4-((prop-2-yn-1-yloxy)methyl)benzene (**7f**)

Compound **7f** was synthesized according to the General Procedure A in the yield of 86.6%.$^1$H NMR (CDCl$_3$, 400 MHz) $\delta$: 7.35–7.27 (m, 4H, Ph-H), 4.59 (s, 2H, O-CH$_2$-Ph), 4.19–4.18 (m, 2H, O-CH$_2$-alkyne), 2.49 (m, 1H, H of terminal alkyne); $^{13}$C NMR (CDCl$_3$, 100 MHz) $\delta$: 135.8, 133.7, 129.5, 129.4, 128.7, 128.6, 70.7, 57.2. HRMS (ESI). M + H$^+$: $C_{10}H_{10}ClO$, calcd for: 181.0420; found: 181.0419. The data of **7f** were in agreement with the data reported in the literature [27,28].

### 4.1.2.6. 1-Bromo-4-((prop-2-yn-1-yloxy)methyl)benzene (**7g**)

Compound **7g** was synthesized according to the General Procedure A in the yield of 81%. $^1$H NMR (CDCl$_3$, 400 MHz) $\delta$: 7.49 (d, $J$ = 8.0 Hz, 2H, Ph-H), 7.25 (d, $J$ = 8.0 Hz, 2H, Ph-H), 4.57 (s, 2H, O-CH$_2$-Ph), 4.18 (d, $J$ = 1.6 Hz, 2H O-CH$_2$-alkyne), 2.49 (m, 1H,H of terminal alkyne); $^{13}$C NMR (CDCl$_3$, 100 MHz) $\delta$: 136.3, 131.6, 129.7, 121.8, 70.7, 57.2. HRMS (ESI). M + H$^+$: $C_{10}H_{10}BrO$, calcd for: 224.9915; found: 224.9914. The data of **7 g** were in agreement with the data reported in the literature [26,29].

### 4.1.2.7. 1-((Prop-2-yn-1-yloxy)methyl)-4-(trifluoromethyl)benzene (**7h**)

Compound **7h** was synthesized according to the General Procedure A in the yield of 81%. $^1$H NMR (CDCl$_3$, 400 MHz) $\delta$: 7.58 (d, $J$ = 8.4 Hz, 2H, Ph-H), 7.44 (d, $J$ = 8.4 Hz, 2H, Ph-H), 4.65 (s, 2H, O-CH$_2$-Ph), 4.21 (m, 2H, CH$_2$-alkyne), 2.48 (m, 1H, H of terminal alkyne); $^{13}$C NMR (CDCl$_3$, 100 MHz) $\delta$: 141.2, 130.0, 127.7, 125.3, 79.1, 75.0, 70.6, 57.5. The data of **7h** were in agreement with the data reported in the literature [30].

### 4.1.2.8. 1-Nitro-4-((prop-2-yn-1-yloxy)methyl)benzene (**7i**)

Compound **7i** was synthesized according to the General Procedure A in the yield of 60%. $^1$H NMR (CDCl$_3$, 400 MHz) $\delta$: 8.21 (d, $J$ = 8.8 Hz, 2H, H-Ph), 7.53 (d, $J$ = 8.8 Hz, 2H, H-Ph), 4.71 (s, 2H, CH$_2$-Ph), 4.26 (d, $J$ = 2.4 Hz, 2H, O-CH$_2$-alkyne), 2.51 (m, 1H, H of terminal alkyne). The $^1$H NMR data of **7i** were in agreement with the data reported in the literature [31].

### 4.1.3. General procedure B: preparation for compound 8a–8i [26]

Compound **4** (300 mg, 0.8036 mmol), compound **7** (0.9643 mmol), copper sulfate pentahydrate (10 mg, 0.04018 mmol) and sodium ascorbate (16 mg, 0.08036 mmol) were dissolved in 1 ml $N$, $N$-dimethylformamide in vial under N$_2$. The reaction was heated to 100°C for 30 min. The solvent was removed under reduced pressure. The residue was dissolved in 50 ml ethyl acetate and washed with 10 ml saturated aqueous sodium chloride solution. The organic layer was dried over anhydrous sodium sulfate. It was filtered and the filtrate was evaporated under reduced pressure. The residue was purified through silica gel chromatography.

### 4.1.3.1. 1-(1,3,4,6-Tetra-O-acetyl-ᴅ-glucopyranosyl)-4-hydroxymethyl-[1,2,3]-triazole (**8a**)

Compound **8a** was synthesized according to the General Procedure B. Compound **4** (300 mg, 0.8036 mmol), compound **7a** (54 mg, 0.9643 mmol), copper sulfate pentahydrate (10 mg, 0.04018 mmol) and sodium ascorbate (16 mg, 0.08036 mmol) were dissolved in 2 ml $N$,$N$-dimethylformamide in vial under N$_2$. The reaction was heated to 100°C for 30 min under microwave. TLC was used to monitor the reaction ($R_f$ = 0.4, petroleum ether/ethyl acetate = 1/4). When the reaction was over, the solvent was removed under reduced pressure. The residue was dissolved in 50 ml ethyl acetate and washed with 10 ml saturated aqueous sodium chloride solution. The organic layer was dried over anhydrous sodium sulfate. It was filtered and the filtrate was evaporated under reduced pressure. The residue was purified through silica gel chromatography with the eluant of petroleum ether and ethyl acetate (1/4). Compound **8a** was obtained as white solid (300.2 mg, 87%) ($\beta$ : $\alpha$ = 1.7 : 1). Compound **8a** was characterized with $^1$H NMR, $^{13}$C NMR and HRMS. $^1$H NMR (400 MHz, CDCl$_3$) $\beta$: $\delta$: 7.65 (s, 1H, H-triazole), 6.27 (d, $J$ = 8.4 Hz, 1H, anomeric H), 5.89–5.84 (m, 1H, CH), 5.36–5.20 (m, 1H, CH), 4.84 (s, 2H, triazole-CH$_2$),4.76–4.71 (m, 1H, CH), 4.47–4.39 (m, 1H, CH), 4.29–4.12 (m, 2H, CH$_2$-OAc), 2.49 (brs, 1H, OH), 2.18–2.05 (m, 12H, O-CO-CH$_3$); $\alpha$: $\delta$: 7.69 (s, 1H, H-triazole), 6.45 (d, $J$ = 3.6 Hz, 1H, anomeric H), 6.04–5.99 (m, 1H, CH), 5.33–5.20

(m, 1H, CH), 4.84 (s, 2H, triazole-CH$_2$), 4.76–4.71 (m, 1H, CH), 4.47–4.39 (m, 1H, CH), 4.29–4.12 (m, 2H, CH$_2$-OAc), 2.49 (brs, 1H, OH), 2.18–2.05 (m, 12H, O-CO-CH$_3$). $^{13}$C NMR (100 MHz, CDCl$_3$) $\alpha + \beta$ $\delta$: 170.6, 170.0, 169.6, 169.3, 169.3, 168.2, 167.9, 148.1, 147.9, 122.1, 120.8, 91.5, 90.0, 72.8, 72.0, 69.8, 68.7, 68.1, 68.0, 62.7, 61.3, 61.3, 61.0, 56.3, 56.2, 20.8, 20.7, 20.7, 20.6, 20.6, 20.5, 20.4, 20.3. HRMS (ESI): M + H$^+$:C$_{17}$H$_{24}$N$_3$O$_{10}$, calcd for: 430.1462; found: 430.1458.

### 4.1.3.2. 1-(1,3,4,6-Tetra-O-acetyl-D-glucopyranosyl)-4-benzhydryloxymethyl-[1,2,3]-triazole (**8b**)

Compound **8b** was synthesized according to the General Procedure B. Compound **4** (300 mg, 0.8036 mmol), compound **7b** (141 mg, 0.9643 mmol), copper sulfate pentahydrate (10 mg, 0.04018 mmol) and sodium ascorbate (16 mg, 0.08036 mmol) were dissolved in 2 ml N,N-dimethylformamide in vial under N$_2$. The reaction was heated to 100°C for 30 min under microwave. TLC was used to monitor the reaction ($R_f$ = 0.86, petroleum ether/ethyl acetate = 1/1). When the reaction was over, the solvent was removed under reduced pressure. The residue was dissolved in 50 ml ethyl acetate and washed with 10 ml saturated aqueous sodium chloride solution. The organic layer was dried over anhydrous sodium sulfate. It was filtered and the filtrate was evaporated under reduced pressure. The residue was purified through silica gel chromatography with the eluant of petroleum ether and ethyl acetate (2/1). Compound **8b** was obtained as white solid in the yield of (279.7 mg, 67%) ($\beta$ : $\alpha$ = 3.4 : 1). $^1$H NMR (400 MHz, CDCl$_3$) $\beta$ : $\delta$: 7.70 (s, 1H, H-triazole), 7.28–7.22 (m, 5H, Ph-H), 6.25 (d, J = 8.8 Hz, 1H, anomeric H), 5.85–5.79 (t, J = 10 Hz, 9.6 Hz, 1H, CH), 5.20–5.15 (t, J = 10 Hz, 9.6 Hz, 1H, CH), 4.70–4.66 (t, J = 9.6 Hz, 9.6 Hz, 1H, CH), 4.61 (s, 2H, Ph-CH$_2$), 4.48 (s, 2H, triazole-CH$_2$), 4.35–4.31 (m, 1H, CH), 4.11–4.05 (m, 2H, CH$_2$-OAc), 2.03–1.78 (m, 12H, O-COCH$_3$); $\alpha$: $\delta$: 7.70 (s, 1H, H-triazole), 7.28–7.22 (m, 5H, Ph-H), 6.34 (d, J = 2.8 Hz, 1H, anomeric H), 5.97–5.91 (t, J = 10.4 Hz, 10.4 Hz, 1H, CH), 5.28–5.21 (t, J = 10 Hz, 9.6 Hz,1H, CH), 4.70–4.66 (t, J = 9.6 Hz, 9.6 Hz,1H, CH), 4.60 (s, 2H, Ph-CH$_2$), 4.48 (s, 2H, triazole-CH$_2$), 4.22 (d, J = 9.6 Hz,1 H), 4.11–4.05 (m, 2H, CH$_2$-OAc), 2.03–1.78 (m, 12H, O-COCH$_3$). $^{13}$C NMR (100 MHz, CDCl$_3$) $\alpha + \beta$ $\delta$: 170.5, 169.8, 169.6, 169.3, 169.1, 169.0, 167.7, 145.5, 137.6, 129.7, 128.5, 127.9, 127.8, 124.8, 122.7, 121.6, 91.6, 90.0, 73.0, 72.4, 72.3, 72.1, 69.8, 68.8, 68.2, 68.0, 63.5, 63.4, 62.7, 61.4, 61.1, 20.7, 20.5, 20.5, 20.3, 20.2. HRMS: M + H$^+$: C$_{24}$H$_{30}$N$_3$O$_{10}$, calcd for: 520.1931; found: 520.1931.

### 4.1.3.3. 1-(1,3,4,6-Tetra-O-acetyl-D-glucopyranosyl)-4-(4-methylbenzene)hydryloxymethyl-[1,2,3]-triazole (**8c**)

Compound **8c** was synthesized according to the General Procedure B. Compound **4** (300 mg, 0.8036 mmol), compound **7c** (154.5 mg, 0.9643 mmol), copper sulfate pentahydrate (10 mg, 0.04018 mmol) and sodium ascorbate (16 mg, 0.08036 mmol) were dissolved in 2 ml N,N-dimethylformamide in vial under N$_2$. The reaction was heated to 100°C for 30 min under microwave. TLC was used to monitor the reaction ($R_f$ = 0.23, petroleum ether/ethyl acetate = 2/1). When the reaction was over, the solvent was removed under reduced pressure. The residue was dissolved in 50 ml ethyl acetate and washed with 10 ml saturated aqueous sodium chloride solution. The organic layer was dried over anhydrous sodium sulfate. It was filtered and the filtrate was evaporated under reduced pressure. The residue was purified through silica gel chromatography with the eluant of petroleum ether and ethyl acetate (2/1). Compound **8c** was obtained as white solid (391 mg, 91.2%) ($\beta$ : $\alpha$ = 3 : 1). $^1$H NMR (400 MHz, CDCl$_3$) $\beta$ : $\delta$: 7.58 (s, 1H, H-triazole), 7.23 (d, J = 8.0 Hz, 2H, Ph-H), 7.16 (d, J = 8.0 Hz, 2H, Ph-H), 6.19 (d, J = 8.8 Hz, 1H, anomeric H), 5.81–5.76 (m, CH, 1H), 5.31–5.15 (m, CH,1H), 4.70–4.67 (m, CH, 1H), 4.65 (s, Ph-CH$_2$, 2H), 4.52 (s, triazole-CH$_2$, 2H), 4.42–4.34 (m, CH, 1H), 4.24–3.99 (m, CH$_2$-OAc, 2H), 2.35 (s, CH$_3$-Ph,3H), 2.11–1.86 (m, O-COCH$_3$, 12H); $\alpha$: $\delta$ 7.64 (s, 1H, H-triazole), 7.23 (d, J = 8.0 Hz, 2H, Ph-H), 7.16 (d, J = 8.0 Hz, 2H, Ph-H), 6.38 (d, J = 3.6 Hz, 1H, anomeric H), 5.97–5.94 (m, CH, 1H), 5.31–5.15 (m, CH, 1H), 4.70–4.67 (m, CH, 1H), 4.64 (s, Ph-CH$_2$, 2H), 4.50 (s, triazole-CH$_2$, 2H), 4.42–4.34 (m, CH, 1H), 4.24–3.99 (m, CH$_2$-OAc, 2H), 2.35 (s, CH$_3$-Ph, 3H), 2.11–1.86 (m, O-COCH$_3$, 12H); $^{13}$C NMR (100 MHz, CDCl$_3$) $\alpha + \beta$ $\delta$ : 170.6, 170.5, 169.8, 169.6, 169.0, 168.1, 167.7, 145.8, 145.6, 137.6, 137.5, 134.5, 134.4, 129.1, 129.0, 128.1, 127.9, 122.7, 121.5, 91.6, 90.0, 72.9, 72.3, 72.1, 72.0, 69.8, 68.7, 68.1, 67.9, 63.3, 63.2, 62.55, 61.3, 61.2, 61.0, 21.1, 20.7, 20.6, 20.5, 20.5, 20.3, 20.2. HRMS: M + H$^+$: C$_{25}$H$_{32}$N$_3$O$_{10}$, calcd for: 534.2088; found: 534.2090.

### 4.1.3.4. 1-(1,3,4,6-Tetra-O-acetyl-D-glucopyranosyl)-4-(4-methoxybenzene)hydryloxymethyl-[1,2,3]-triazole (**8d**)

Compound **8d** was synthesized according to the General Procedure B. Compound **4** (300 mg, 0.8036 mmol), compound **7d** (169.9 mg, 0.9643 mmol), copper sulfate pentahydrate (10 mg, 0.04018 mmol) and sodium ascorbate (16 mg, 0.08036 mmol) were dissolved in 2 ml N,N-dimethylformamide in vial under N$_2$. The reaction was heated to 100°C for 30 min under microwave. TLC was used to monitor the reaction ($R_f$ =

0.51, petroleum ether/ethyl acetate = 1/1). When the reaction was over, the solvent was removed under reduced pressure. The residue was dissolved in 50 ml ethyl acetate and washed with 10 ml saturated aqueous sodium chloride solution. The organic layer was dried over anhydrous sodium sulfate. It was filtered and the filtrate was evaporated under reduced pressure. The residue was purified through silica gel chromatography with the eluant of petroleum ether and ethyl acetate (2/1). Compound **8d** was obtained as white solid (301.1 mg, 68.2%) ($\beta : \alpha$ = 2.7 : 1). $^1$H NMR (400 MHz, CDCl$_3$) $\beta$: $\delta$: 7.58 (s, 1H, H-triazole), 7.28 (t, Ph-H, 2H), 6.89 (t, Ph-H, 2H), 6.19 (d, $J$ = 3.6 Hz, 1H, anomeric H), 5.81–5.76 (m, CH, 1H), 5.31–5.15 (m, CH, 1H), 4.70–4.68 (m, CH, 1H), 4.64 (s, Ph-CH$_2$, 2H), 4.49 (s, triazole-CH$_2$, 2H), 4.42–4.34 (m, CH, 1H), 4.24–4.04 (m, 2H), 3.81(s, CH$_3$O, 3H), 2.12–1.86 (m, O-COCH$_3$, 12H); $\alpha$: $\delta$: 7.64 (s, 1H, H-triazole), 7.28 (t, Ph-H, 2H), 6.89 (t, Ph-H, 2H), 6.38 (d, $J$ = 3.6 Hz, 1H, anomeric H), 6.00–5.94 (m, CH, 1H), 5.31–5.15 (m, 1H), 4.70–4.68 (m, CH, 1H), 4.64 (s, Ph-CH$_2$, 2H), 4.49 (s, triazole-CH$_2$, 2H), 4.42–4.34 (m, CH, 1H), 4.24–4.04 (m, 2H), 3.81 (s, CH$_3$O, 3H), 2.12–1.86 (m, O-COCH$_3$,12H); $^{13}$C NMR (100 MHz, CDCl$_3$) $\alpha + \beta$ $\delta$: 170.5, 169.9, 169.6, 169.3, 169.1, 168.1, 167.7, 159.4, 159.2, 145.8, 145.5, 129.6, 129.6, 129.5, 129.4, 122.7, 121.6, 113.8, 113.7, 91.6, 90.0, 72.9, 72.1, 71.9, 72.1, 69.8, 68.7, 68.1, 67.9, 63.1, 63.0, 62.6, 61.3, 61.2, 61.0, 55.2, 21.1, 20.7, 20.6, 20.5, 20.3, 20.2. HRMS: M + H$^+$: C$_{25}$H$_{32}$N$_3$O$_{11}$, calcd for: 550.2037; found: 550.2028.

### 4.1.3.5. 1-(1,3,4,6-Tetra-O-acetyl-D-glucopyranosyl)-4-(4-fluorobenzene)hydryloxymethyl-[1,2,3]-triazole (8e)

Compound **8e** was synthesized according to the General Procedure B. Compound **4** (300 mg, 0.8036 mmol), compound **7e** (158.3 mg, 0.9643 mmol), copper sulfate pentahydrate (10 mg, 0.04018 mmol) and sodium ascorbate (16 mg, 0.08036 mmol) were dissolved in 2 ml N,N-dimethylformamide in vial under N$_2$. The reaction was heated to 100°C for 30 min under microwave. TLC was used to monitor the reaction ($R_f$ = 0.41, petroleum ether/ethyl acetate = 1/1). When the reaction was over, the solvent was removed under reduced pressure. The residue was dissolved in 50 ml ethyl acetate and washed with 10 ml saturated aqueous sodium chloride solution. The organic layer was dried over anhydrous sodium sulfate. It was filtered and the filtrate was evaporated under reduced pressure. The residue was purified through silica gel chromatography with the eluant of petroleum ether and ethyl acetate (2/1). Compound **8e** was obtained as white solid (372.2 mg, 86.17%) ($\beta : \alpha$ = 6.7 : 1). $^1$H NMR (400 MHz, CDCl$_3$) $\beta$ : $\delta$: 7.60 (s, 1H, H-triazole), 7.36–7.29 (m, 4H, Ph-H), 6.20 (d, $J$ = 8.8 Hz, 1H, anomeric H), 5.81–5.77 (m, CH, 1H), 5.25–5.20 (m, CH, 1H), 4.71–4.66 (m, Ph-CH$_2$ and CH, 3H), 4.57 (s, triazole-CH$_2$, 2H), 4.42–4.38 (m, 1H, CH), 4.24–4.06 (m, 2H, CH$_2$-OAc), 2.12–1.79 (m, O-COCH$_3$, 12H). $\alpha$: $\delta$: 7.65 (s, 1H, H-triazole),7.36–7.29 (m, 4H, Ph-H), 6.38 (d, $J$ = 3.6 Hz, 1H, anomeric H), 6.00–5.95 (m, CH, 1H), 5.25–5.20 (m, CH, 1H), 4.71–4.66 (m, Ph-CH$_2$ and CH, 3H), 4.56 (s, triazole-CH$_2$, 2H), 4.36–4.34 (m, 1H, CH), 4.24–4.06 (m, 2H, CH$_2$-OAc), 2.12–1.79 (m, O-COCH$_3$,12H). $^{13}$C NMR (100 MHz, CDCl$_3$) $\alpha + \beta$: $\delta$: 170.5, 169.6, 169.1, 168.1, 145.5, 137.5, 128.4, 128.4, 127.9, 127.8, 122.8, 121.6, 91.6, 90.0, 72.9, 72.4, 72.2, 72.1, 69.8, 68.7, 68.0, 63.4, 63.4, 62.6, 61.3, 61.0. 20.7, 20.6, 20.5, 20.3, 20.2. HRMS: M + H$^+$: C$_{24}$H$_{29}$FN$_3$O$_{10}$, calcd for: 538.1837; found: 538.1848.

### 4.1.3.6. 1-(1,3,4,6-Tetra-O-acetyl-D-glucopyranosyl)-4-(4-chlorobenzene)hydryloxymethyl-[1,2,3]-triazole (8f)

Compound **8f** was synthesized according to the General Procedure B. Compound **4** (300 mg, 0.8036 mmol), compound **7f** (174.2 mg, 0.9643 mmol), copper sulfate pentahydrate (10 mg, 0.04018 mmol) and sodium ascorbate (16 mg, 0.08036 mmol) were dissolved in 2 ml N,N-dimethylformamide in vial under N$_2$. The reaction was heated to 100°C for 30 min under microwave. TLC was used to monitor the reaction ($R_f$ = 0.57, petroleum ether/ethyl acetate = 1/1). When the reaction was over, the solvent was removed under reduced pressure. The residue was dissolved in 50 ml ethyl acetate and washed with 10 ml saturated aqueous sodium chloride solution. The organic layer was dried over anhydrous sodium sulfate. It was filtered and the filtrate was evaporated under reduced pressure. The residue was purified by prepared TLC with the eluant of petroleum ether/ethyl acetate (1/2). Compound **8f** was obtained as white solid (240 mg, 53.9%) ($\beta : \alpha$ = 4.8 : 1). $^1$H NMR (400 MHz, CDCl$_3$) $\beta$: $\delta$: 7.55 (s, 1H, H-triazole), 7.25 (m, 4H, Ph-H), 6.15 (d, $J$ = 8.4 Hz, 1H, anomeric H), 5.85–5.72 (m, 1H, CH), 5.26–5.13 (m, 1H, CH), 4.68–4.63 (m, Ph-CH$_2$ and CH, 3H), 4.49 (s, 2H), 4.38–4.27 (m, 1H), 4.15–4.02 (m, 2H), 2.10–1.84 (m, O-COCH$_3$, 12H); $\alpha$: $\delta$: 7.61 (s, 1H, H-triazole), 7.25 (m, 4H, Ph-H), 6.35 (d, $J$ = 3.6 Hz, 1H, anomeric H), 6.02–5.97 (m, 1H, CH), 5.28–5.24 (m, 1H, CH), 4.74–4.65 (m, Ph-CH$_2$ and CH, 3H), 4.55 (s, 2H), 4.45–4.38 (m, 1H), 4.25–4.08 (m, 2H), 2.18–1.86 (m, O-COCH$_3$, 12H); $^{13}$C NMR (100 MHz, CDCl$_3$) $\alpha + \beta$: $\delta$: 170.2, 169.3, 168.8, 167.8, 145.0, 135.9, 133.4, 129.0, 128.9, 128.4, 122.6, 121.5, 91.5, 89.9, 72.9, 72.1, 71.5, 71.4, 69.8, 69.2, 68.7, 68.1, 67.9,

63.4, 62.7, 61.8, 61.3, 61.1, 20.9, 20.8, 20.7, 20.5, 20.4. HRMS: M + H$^+$: $C_{24}H_{29}ClN_3O_{10}$, calcd for: 554.1541; found: 554.1548; M + 2 + H$^+$: calcd for: 556.1525; found: 556.1529.

### 4.1.3.7. 1-(1,3,4,6-Tetra-O-acetyl-D-glucopyranosyl)-4-(4-bromobenzene)hydryloxymethyl-[1,2,3]-triazole (8g)

Compound **8g** was synthesized according to the General Procedure B. Compound **4** (300 mg, 0.8036 mmol), compound **7g** (206.5 mg, 0.9643 mmol), copper sulfate pentahydrate (10 mg, 0.04018 mmol) and sodium ascorbate (16 mg, 0.08036 mmol) were dissolved in 2 ml N,N-dimethylformamide in vial under N$_2$. The reaction was heated to 100°C for 30 min under microwave. TLC was used to monitor the reaction ($R_f$ = 0.58, petroleum ether/ethyl acetate = 1/1). When the reaction was over, the solvent was removed under reduced pressure. The residue was dissolved in 50 ml ethyl acetate and washed with 10 ml saturated aqueous sodium chloride solution. The organic layer was dried over anhydrous sodium sulfate. It was filtered and the filtrate was evaporated under reduced pressure. The residue was purified through silica gel chromatography with the eluant of petroleum ether/ethyl acetate (1/2). Compound **8g** was obtained as white solid (437.6 mg, 91%)($\beta$ : $\alpha$ = 3 : 1). $^1$H NMR (400 MHz, CDCl$_3$) $\beta$: $\delta$: 7.55 (s, 1H, H-triazole), 7.44 (d, J = 7.4 Hz, 2H, Ph-H), 7.21 (d, J = 7.4 Hz, 2H, Ph-H), 6.16 (d, J = 8.4 Hz, 1H, anomeric H), 5.77–5.72 (m, 1H, CH), 5.29–5.18 (m, 1H, CH), 4.69–4.63 (m, 3H, Ph-CH$_2$ and CH), 4.48 (s, 2H, triazole-CH$_2$), 4.39–4.33 (m, 1H, CH), 4.20–4.02 (m, 2H, CH$_2$-OAc), 2.10–1.68 (m, O-COCH$_3$, 12H). $\alpha$: $\delta$: 7.61(s, 1H, H-triazole), 7.44 (d, J = 7.4 Hz, 2H, Ph-H), 7.21 (d, J = 7.4 Hz, 2H, Ph-H), 6.35 (d, J = 3.6 Hz, 1H, anomeric H), 5.96–5.91 (m, 1H, CH), 5.29–5.18 (m, 1H, CH), 4.69–4.63 (m, 3H, Ph-CH$_2$ and CH), 4.47 (s, 2H, triazole-CH$_2$), 4.39–4.33 (m, 1H, CH), 4.20–4.02 (m, 2H, CH$_2$-OAc), 2.10–1.68 (m, 12H, O-COCH$_3$). $^{13}$C NMR (100 MHz, CDCl$_3$) $\alpha$ + $\beta$: $\delta$: 170.2, 169.3, 168.8, 167.8, 145.0, 131.3, 129.3, 122.5, 121.5, 91.5, 89.9, 72.9, 72.1, 71.5, 71.4, 69.8, 68.7, 68.1, 67.9, 63.4, 62.7, 61.3, 61.1, 20.8, 20.7, 20.5, 20.4. HRMS: M + H$^+$: $C_{24}H_{29}BrN_3O_{10}$, calcd for: 598.1036; found: 598.1044; M + 2 + H$^+$: calcd for: 600.1020; found: 600.1027.

### 4.1.3.8. 1-(1,3,4,6-Tetra-O-acetyl-D-glucopyranosyl)-4-(4-trifluoromethylbenzene)hydryloxymethyl-[1,2,3]-triazole (8h)

Compound **8h** was synthesized according to the General Procedure B. Compound **4** (300 mg, 0.8036 mmol), compound **7h** (217 mg, 0.9643 mmol), copper sulfate pentahydrate (10 mg, 0.04018 mmol) and sodium ascorbate (16 mg, 0.08036 mmol) were dissolved in 2 ml N,N-dimethylformamide in vial under N$_2$. The reaction was heated to 100°C for 30 min under microwave. TLC was used to monitor the reaction ($R_f$ = 0.55, petroleum ether/ethyl acetate = 1/1). When the reaction was over, the solvent was removed under reduced pressure. The residue was dissolved in 50 ml ethyl acetate and washed with 10 ml saturated aqueous sodium chloride solution. The organic layer was dried over anhydrous sodium sulfate. It was filtered and the filtrate was evaporated under reduced pressure. The residue was purified through silica gel chromatography with the eluant of petroleum ether/ethyl acetate (1/2). Compound **8h** was obtained as white solid (267.6 mg, 56.69%) ($\beta$ : $\alpha$ = 7.7 : 1). $^1$H NMR (400 MHz, CDCl$_3$) $\beta$: $\delta$: 7.58 (m, 3H, Ph-H and H-triazole), 7.43 (d, J = 8.0 Hz, 2H, Ph-H), 6.17 (d, J = 8.8 Hz, 1H, anomeric H), 5.97–5.92 (m, 1H, CH), 5.23–5.19 (m, 1H, CH), 4.70–4.67 (m, 3H, CH$_2$-Ph and CH), 4.60 (s, 2H, triazole-CH$_2$), 4.39–4.38 (m, 1H, CH), 4.17–4.04 (m, 2H, CH$_2$-OAc), 2.11–1.85 (m, 12H, O-COCH$_3$). $\alpha$: $\delta$: 7.58 (m, 3H, Ph-H and H-triazole), 7.43 (d, J = 8.0 Hz, 2H, Ph-H), 6.37 (d, J = 3.6 Hz, 1H, anomeric H), 5.97–5.92 (m, 1H, CH), 5.50–5.46 (m, 1H, CH), 4.70–4.67 (m, 3H, CH$_2$-Ph and CH), 4.60 (s, 2H, triazole-CH$_2$), 4.39–4.38 (m, 1H, CH), 4.17–4.04 (m, 2H, CH$_2$-OAc), 2.11–1.85 (m, 12H, O-COCH$_3$). $^{13}$C NMR (100 MHz, CDCl$_3$) $\beta$: $\delta$: 170.2, 169.3, 168.8, 167.8, 144.9, 141.5, 127.6, 125.2, 122.6, 91.5, 89.9, 73.0, 72.1, 71.4, 67.9, 63.6, 62.7, 61.3, 20.9, 20.7, 20.4. HRMS: M + H$^+$: $C_{25}H_{29}F_3N_3O_{10}$, calcd for: 588.1805; found: 588.1813.

### 4.1.3.9. 1-(1,3,4,6-Tetra-O-acetyl-D-glucopyranosyl)-4-(4-nitrobenzene)hydryloxymethyl-[1,2,3]-triazole (8i)

Compound **8i** was synthesized according to the General Procedure B. Compound **4** (300 mg, 0.8036 mmol), compound **7i** (184.4 mg, 0.9643 mmol), copper sulfate pentahydrate (10 mg, 0.04018 mmol) and sodium ascorbate (16 mg, 0.08036 mmol) were dissolved in 2 ml N,N-dimethylformamide in vial under N$_2$. The reaction was heated to 100°C for 30 min under microwave. TLC was used to monitor the reaction ($R_f$ = 0.49, petroleum ether/ethyl acetate = 1/1). When the reaction was over, the solvent was removed under reduced pressure. The residue was dissolved in 50 ml ethyl acetate and washed with 10 ml saturated aqueous sodium chloride solution. The organic layer was dried over anhydrous sodium sulfate. It was filtered and the filtrate was evaporated under

reduced pressure. The residue was purified through silica gel chromatography with the eluant of petroleum ether/ethyl acetate (1/2). Compound **8i** was obtained as yellow solid (403.3 mg, 88.9%) ($\beta$ : $\alpha = 6.7 : 1$). $^1$H NMR (400 MHz, CDCl$_3$) $\beta$: $\delta$: 8.21 (d, $J = 8.4$ Hz, 2H, Ph-H), 7.64 (s, 1H, H-triazole), 7.52 (d, $J = 8.4$ Hz, 2H, Ph-H), 6.21 (d, $J = 8.8$ Hz, 1H, anomeric H), 5.79–5.74 (m, 1H, CH), 5.23–5.21 (m, 1H,CH), 4.75–4.70 (m, 3H, CH$_2$-Ph and CH), 4.66 (s, 2H, triazole-CH$_2$), 4.42–4.37 (m, 1H, CH), 4.19–4.05 (m, 2H,CH$_2$-OAc), 2.11–1.83 (m, 12H, O-COCH$_3$). $\alpha$: $\delta$ : 8.21 (d, $J = 8.4$ Hz, 2H, Ph-H), 7.69 (s, 1H, H-triazole), 7.52 (d, $J = 8.4$ Hz, 2H, Ph-H), 6.40 (d, $J = 3.6$ Hz, 1H, anomeric H), 6.00–5.95 (m, 1H, CH), 5.32–5.27 (m, 1H,CH), 4.75–4.70 (m, 3H, CH$_2$-Ph and CH), 4.65 (s, 2H, triazole-CH$_2$), 4.42–4.37 (m, 1H, CH), 4.19–4.05 (m, 2H,CH$_2$-OAc), 2.11–1.83 (m, 12H, O-COCH$_3$). $^{13}$C NMR (100 MHz, CDCl$_3$) $\alpha + \beta$: $\delta$ : 170.5, 169.6, 169.1, 168.1, 147.5, 145.3, 144.9, 129.0, 127.9, 127.8, 123.7, 123.6, 122.8, 121.9, 91.6, 90.1, 73.0, 72.1, 70.9, 70.8, 69.9, 68.8, 68.1, 68.0, 63.9, 62.8, 61.4, 61.2, 20.7, 20.6, 20.5, 20.3, 20.2. HRMS: M + H$^+$: C$_{24}$H$_{29}$N$_4$O$_{12}$, calcd for: 565.1782; found: 565.1784; M + Na$^+$: C$_{24}$H$_{28}$N$_4$NaO$_{12}$, calcd for: 587.1601; found: 587.1600.

### 4.1.4. 1-(D-glucopyranosyl)-4-(4-methoxybenzene)hydryloxymethyl-[1,2,3]-triazole (**9d**)

Compound **9d** was prepared according to the literature [32]. Compound **8d** (200 mg, 0.3640 mmol) was dissolved in 8 ml mixture solvent of methanol, water and NEt$_3$ (the ratio of volume was as followed: $V_{MeOH}$ : o$V_{water}$ : $V_{triethylamine} = 5 : 4 : 1$). It was heated to reflux for 4 h. TLC was used to monitor the reaction ($R_f = 0.32$, CH$_2$Cl$_2$/MeOH = 40/1). When the reaction was over, the solvent was removed under reduced pressure. The residue was purified by preparation TLC with the eluant of dichloromethane and methanol (10/1). Compound **9d** was obtained as colourless syrup (77.1 mg, 55.5%). $^1$H NMR (400 MHz, CDCl$_3$) ($\beta$ : $\alpha = 0.9 : 1$) $\beta$ : $\delta$ 7.85 (s, 1H, H-triazole), 7.14 (d, $J = 8.6$ Hz, 2H, Ph-H), 6.75 (d, $J = 8.4$ Hz, 2H, Ph-H), 5.14 (d, $J = 8.0$ Hz, 1H, anomeric H), 4.90 (s, 1H, CH), 4.61 (d, $J = 8.0$ Hz, 1H, CH), 4.47 (s, 2H, CH$_2$-Ph), 4.37 (s, 2H, triazole-CH$_2$), 4.20 (m, 1H, CH), 3.95–3.74 (m, 4H, OH), 3.67 (s, 3H, CH$_3$O), 3.52 (s, 1H, CH), 3.05 (m, 2H, CH$_2$-OH). $\alpha$: $\delta$ 7.72 (s, 1H, H-triazole), 7.14 (d, $J = 8.6$ Hz, 2H, Ph-H), 6.75 (d, $J = 8.4$ Hz, 2H, Ph-H), 5.89 (s, 1H, anomeric OH), 5.22 (d, $J = 6.8$ Hz, 1H, anomeric H), 4.80 (s, 1H, CH), 4.54 (m, 1H, CH), 4.47 (s, 2H, CH$_2$-Ph), 4.37 (s, 2H, triazole-CH$_2$), 4.11–4.07 (t, $J = 8.0$ Hz, 1H, CH), 3.95–3.74 (m, 3H, OH), 3.67 (s, 3H, CH$_3$O), 3.52 (s, 1H, CH), 3.03 (m, 2H, CH$_2$-OH). $^{13}$C NMR (100 MHz, CDCl$_3$) $\alpha + \beta$: $\delta$ : 158.3, 143.7, 143.3, 128.8, 128.7, 128.5, 128.4, 122.3, 112.8, 93.7, 90.2, 82.6, 79.5, 75.1, 71.2, 70.6, 69.6, 66.0, 61.9, 61.8, 60.4, 59.8, 59.4, 56.6, 54.2. HRMS: M + H$^+$: C$_{17}$H$_{24}$N$_3$O$_7$, calcd for: 382.1609; found: 382.1610.

## 4.2. Cell lines and cell culture

The human colon cancer HCT-116 cells and human embryonic kidney cells 293T were purchased from the American Type Culture Collection (ATCC). HCT-116 cells were cultured in McCoy's 5a Modified Medium (Gibco, USA). 293T cells were cultured in Dulbecco's Modified Eagle Medium (Gibco, USA). Both media contain 10% fetal bovine serum (Gibco, USA) and 1% penicillin–streptomycin (Gibco, USA). Cells were maintained in incubator (Thermo Fisher, USA) supplied with 5% CO$_2$ at 37°C. Only cells in logarithmic phase were used in all following experiments.

## 4.3. MTS assay

MTS assay was used to evaluate the cytotoxicity of the compounds, 5-Fu, **8a–8i** and **9d** in HCT 116 and 293T cells. In brief, cells were seeded into 96-well plate (Corning, USA) with a density of 3000 cells/well for 24 h, and then treated with tested compounds at various concentrations for 72 h. MTS solution was added to each well for 4 h, and absorbance was measured at 495 nm by using Varioskan Flash Multimode Reader (Thermo, USA). The values of IC$_{50}$ were calculated by GraphPad Prism Software 7.0.

## 4.4. Live cell imaging

Cells were seeded into 24-well plate ($1 \times 10^4$ cells well$^{-1}$) for 24 h. Different concentrations of compound **8d** were added to treat cells. Time-lapse photography of cells was recorded by Incucyte system. The confluences relative to total well area were calculated using a module of the cell confluence area analysing.

## 4.5. Colony formation assay

HCT-116 cells were seeded in 6-well plates at a density of 300 cells per well and cultured overnight, and then treated with various concentrations of compound **8d** for 24 h. Cells were continuously cultured for 9 days after drug withdrawal. Finally, 1% crystal violet was used to stain cells after fixing with 4% paraformaldehyde, and the visible colonies were recorded by the microscope and counted manually using imageJ.

## 4.6. Hoechst 33258 staining

HCT-116 cells were seeded in coverslips for 24 h, and then compound **8d** was incubated at a concentration of 5 µM for 48 or 72 h. Cells were incubated with Hoechst 33258 (Sigma, USA) in the dark at room temperature for 30 min after being fixed with 4% paraformaldehyde. The nuclear morphology of apoptosis was observed by fluorescence microscope (Olympus, Japan).

## 4.7. TUNEL assay

TdT-UTP nick end-labelling (TUNEL) assay was used to detect the apoptotic cells by *in situ* cell death detection kit (Roche, Germany) according to the manufacturer's instruction. Briefly, HCT-116 cells were seeded on coverslips and cultured overnight, following treatment with compound **8d** (5 µM) for 72 h, then fixed with 4% paraformaldehyde for 30 min and permeabilized with 0.1% Triton X-100 for 2 min on ice. TUNEL reaction mixture was applied to the cells in an atmosphere for 1 h at 37°C in the dark, following with converter-POD and DAB substrate treatment. Samples were detected by a microscope (Olympus, Japan).

## 4.8. Annexin V staining in living cells

Cells that seed on coverslips were treated with compound **8d** (5 µM) for 72 h. Annexin V-FITC was used to target exposed phosphatidylserine which is a marker of early apoptosis. The nucleus of living cells was labelled by Hoechst 33258. The cells were observed by fluorescence microscope (Olympus, Japan).

## 4.9. Annexin V/PI double staining

Cells were seeded in 6-well plates with a density of $5 \times 10^5$ cells per well and then treated with various concentrations of compound **8d** (0 µM, 2.5 µM and 5 µM) for 48 h. Cells were collected and stained with annexin V and PI for 20 min, and then detected by flow cytometer (BD, USA) to analyse the apoptotic cell distribution. Living cells (annexin $V^-/PI^-$), early apoptotic cells (annexin $V^+/PI^-$), late apoptotic cells (annexin $V^+/PI^+$) and necrotic cells (annexin $V^-/PI^+$) were calculated in the Cell Quest software (BD, USA).

## 4.10. Statistical analysis

Experiments were repeated at least three times. Data are presented as mean and s.d. The Student $T$-test or one-way ANOVA were used to assess differences in statistical analysis by GraphPad Prism Software. $p < 0.05$ represents statistically significant.

Data accessibility. The datasets supporting this article have been uploaded as part of the electronic supplementary material.

Authors' contributions. B.F. contributed to synthesis of the target compounds and wrote part of the article. Y.L. contributed to most anti-cancer biological activities tests and wrote part of article. S.P., X.W. and J.H. contributed to the partial biological activities tests. L.L. contributed to the synthesis of partial intermediates. C.X. contributed to revise the biological activities tests. D.L. revised the whole manuscript. Q.C. contributed to design the idea of manuscript.

Competing interests. We have no competing interests.

Funding. The source of funding for each author: Natural Science Foundation of Xiaogan, China (XGKJ2019010046), the Hubei University Excellent Young and Middle-Aged Science and Technology Innovation Team Project (grant no. T201816) and Natural Science Foundation of Hubei Province, China (2014CFB570) to B.F., National Natural Science Foundation of China (21503075) to C.X., National Natural Science Foundation of China (81502022) to Q.C. and Fundamental Research Fund for the Sun Yat-sen University (16ykpy35) to X.W.

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
