## [Peer Review File · Royal Society Open Science]

Review History

RSOS-201642.R0 (Original submission)

Review form: Reviewer 1

Is the manuscript scientifically sound in its present form?

Yes

Are the interpretations and conclusions justified by the results?

Yes

Is the language acceptable?

Yes

Do you have any ethical concerns with this paper?

No

Have you any concerns about statistical analyses in this paper?

No

Recommendation?

Accept with minor revision (please list in comments)

Comments to the Author(s)

The manuscript RSOS-201642 is devoted to the synthesis of novel Glucopyranosyl conjugated benzyl derivatives as potential anticancer agents and can be interesting to the specialists working in this field. The reviewed article is interesting and the theme of the article meets the scope of the journal. Work is performed at sufficient scientific level and has good quality; the results of investigation are professionally interpreted. However, it needs major revision before publication. To improve the quality and perception of the manuscript I would suggest paying attention to following comments:

- 1) All ¹³C NMR should be to 1 dp, but not to 2 dp (experimental part).
- 2) Moderate English changes required. There are grammar and orthographical errors in the manuscript, which should be corrected.

My decision is minor revision.

Review form: Reviewer 2

Is the manuscript scientifically sound in its present form?

No

Are the interpretations and conclusions justified by the results?

No

Is the language acceptable?

No

Do you have any ethical concerns with this paper?

No

Have you any concerns about statistical analyses in this paper?

No

Recommendation?

Major revision is needed (please make suggestions in comments)

Comments to the Author(s)

The paper of Fu and co-workers deals with the synthesis and cytotoxic activity of glucopyranosyl bearing benzyl derivatives. The main issue with this paper is that the synthesized compounds were tested as mixtures of anomers (in different α, β ratios) and not as pure compounds. Because of this, it is almost impossible to compare the cytotoxic activities of the different derivatives and to conclude about possible structure-activity relationships. I suggest that the authors isolate both the anomers and test them independently. Furthermore, I have found a lot of typographical errors and the manuscript bears the stamp of a non-native English speaker. I suggest the authors ask a Native-English speaker to carefully check their manuscript. I have also other corrections that are written below:

- 1) Title: Change "for" to "against"

- 2) Introduction, page 1: Change "5-fluorouracil" to "5-Fluorouracil"
- 3) Introduction, page 1: Change "to limit" for "limiting"
- 4) Introduction, page 1: Change ".1" to ".[1]"
- 5) Introduction, page 1: You write that "Benzyl and its derivatives are important pharmacophore in many anticancer compounds". So I suggest you add a Figure showing the structures of some of these compounds.
- 6) Introduction, page 2: Change "by the "click chemistry"" to "by "click chemistry" (2 times)
- 7) Introduction, page 2: Change "compling" to "coupling"
- 8) Introduction, page 2: Change "conjugate" to "conjugate"; "conjugating" to "conjugating"; "conjuate" to "conjugate"
- 9) Introduction, page 2: At the end of the introduction section, I would be clearer saying that your target compounds were the one with modification at C2.
- 10) Results and discussion, page 2: "8a-i" must be bolded; "4" must be bolded
- 11) Results and discussion, page 2: Change "itermediate" to "intermediate"; "avaliabe" to "available"; "chromatagraphy" to "chromatography"
- 12) Results and discussion, page 2: Change "by the deacetylation" to "by deacetylation"
- 13) Results and discussion, page 2: You write that "All the products were purified by silica gel chromatography". I would add by which methods the compounds were characterized.
- 14) Scheme 1, page 3: For compounds 3 and 4, what was the anomeric ratio? Why was only compound 9d deprotected?
- 15) Scheme 1 caption: Change "Conditions of reactions" to "reaction conditions"; "4" must be bolded. Add a "space" between "4" and "h". Delete the "space" between "N," and "N"
- 16) Title of section 3.2: I would rather write "The cytotoxicity of the synthesized compounds against..."
- 17) Section 3.2, page 3: Change "coloreactal" to "colorectal"
- 18) Section 3.2, page 3: Please define "MTS"; Change "via MTS" to "via a MTS". Deleted "the" before "cancer cells"; Change "to HCT-116" to "toward HCT-116"; Change "antiproliferation activity is fairly" to "antiproliferative activity was fairly". Delete "," after "8b-8i"; Change "showed 23 fold" to "showed a 23-fold"
- 19) Table 1: It would have been interesting to test peracetylated glucose as a control. The standard deviations \pm SD must be written with only one significative figures and the values of the IC50 must be rounded up accordingly. For instance, "4.23 \pm 1.98" should be written as "4 \pm 2" and so on for the other values.
- 20) Page 4: Sections 3.3 and 3.4 are identical!!! Please delete one section and changer section numbering accordingly.
- 21) Page 4, section 3.3: Change "aganist" to "against"; "As showed" to "As shown"; "Figure 1A, 1C and 1D" to "Figures 1A, 1C, and 1D"; "8d" must be bolded. Change "Figure 1C &1D" to "Figures 1C and 1D"
- 22) Conclusion: Change "aganist" to "against" (2 times); "selectiity" to "selectivity"; "nomal" to "normal"; "subsitution" to "substitution"; "para-position" to "para position"; "influences" to "influenced"; "theapetuic" to "therapeutic"; "is from inducing" to "was shown to induce"
- 23) Section 5.1: Change "propargyl" to "Propargyl"; Change "and removed" by "and solvents were removed".
- 24) Section 5.1: You write that "purities of the intermediates were established by TLC"!!! It is only possible to assess qualitatively the purity by TLC. The proper way to do it is by HPLC and/or elemental analyses.
- 25) Section 5.1: Change "Thin Layer" to "Thin layer"; "water used was distilled" to "water was distilled"
- 26) Section 5.1: Add a "space" between "400" and "MHz" and between "100" and "MHz" (please correct this anywhere it appears in the manuscript"

- 27) Experimental: Please add the aspect of all compounds. The letter "D" in sugars must be written in small capital letters. The "O" must be italicized in compound names. Delete the additional "space" between "of" and "compound"; add an "s" to "compound 7b-7i"
- 28) Experimental: Change "3-bromopropyne" to "3-Bromopropyne"; "completeion" to "completion"; "silca gel" to "silica gel"
- 29) Experimental: Change "compound 8a □8i" to "compound 8a-8i"; add a "space" between "1" and "mL"; delete the "space" between "N," and "N-dimethylformamide"; change "The organic layers were dried" by "The organic layer was dried"; change "sodium chloride" to "saturated aqueous sodium chloride solution"
- 30) Experimental: Delete the unneeded "spaces" in compound names.
- 31) Section 5.6: Change "the incubated compound 8d" to "then compound 8d was incubated"; change "after fixed" to "after being fixed"
- 32) Section 5.7: Change "following treated" to "following treatment"
- 33) Section 5.9: Add a "space" between "5.9" and "Annexin"
- 34) Supporting information: Change "HR-MS" to HRMS"; α, β anomeric ratios must be written for each compound such as for 9b. For 9b, the α, β anomeric ratio is written but this is an equilibrium. Was the NMR spectrum taken at equilibrium?

Decision letter (RSOS-201642.R0)

Dear Dr Fu:

Title: Synthesis and Pharmacological Characterization of Glucopyranosyl-conjugated Benzyl Derivatives as Novel Selective Cytotoxic Agents for Colon Cancer
 Manuscript ID: RSOS-201642

The editor assigned to your manuscript has now received comments from reviewers. We would like you to revise your paper in accordance with the referee and Subject Editor suggestions which can be found below (not including confidential reports to the Editor). Please note this decision does not guarantee eventual acceptance.

Please submit your revised paper before 07-Nov-2020. Please note that the revision deadline will expire at 00.00am on this date. If we do not hear from you within this time then it will be assumed that the paper has been withdrawn. In exceptional circumstances, extensions may be possible if agreed with the Editorial Office in advance. We do not allow multiple rounds of revision so we urge you to make every effort to fully address all of the comments at this stage. If deemed necessary by the Editors, your manuscript will be sent back to one or more of the original reviewers for assessment. If the original reviewers are not available we may invite new reviewers.

To revise your manuscript, log into <http://mc.manuscriptcentral.com/rsos> and enter your Author Centre, where you will find your manuscript title listed under "Manuscripts with Decisions." Under "Actions," click on "Create a Revision." Your manuscript number has been

appended to denote a revision. Revise your manuscript and upload a new version through your Author Centre.

On behalf of the Subject Editor Professor Anthony Stace and the Associate Editor Dr Andrew Harned.

RSC Associate Editor:

Comments to the Author:

The authors have expressed enthusiasm for the work as a whole, but believe the manuscript could be improved by further editing. In addition, Reviewer 2 has requested that all compounds be tested again as single anomers, rather than the anomeric mixtures that were already tested. It is not clear to me how feasible this will be, or how improved the results would be by doing this. Rather than attempting what would likely be a tedious separation for every compound, I suggest performing this purification/reanalysis of compound 8d (much like the authors already did by testing 9d). I also agree with Reviewer 2 that judging purities by TLC is not sufficient. At the very least, HPLC purities should be reported (I do note that the provided NMR spectra do show the compounds are of relatively high purity).

In addition to the extensive grammatical and typographical corrections suggested by Reviewer 2, I suggest changing the anomer ratios provided in the experimental section so that they all have the same format, e.g. 2.7:1 instead of 100:37. This will make it easier for readers to compare the different ratios.

RSC Subject Editor:

Comments to the Author:

(There are no comments.)

Reviewers' Comments to Author:

Reviewer: 1

Comments to the Author(s)

The manuscript RSOS-201642 is devoted to the synthesis of novel Glucopyranosyl conjugated benzyl derivatives as potential anticancer agents and can be interesting to the specialists working in this field. The reviewed article is interesting and the theme of the article meets the scope of the journal. Work is performed at sufficient scientific level and has good quality; the results of investigation are professionally interpreted. However, it needs major revision before publication. To improve the quality and perception of the manuscript I would suggest paying attention to following comments:

- 1) All ¹³C NMR should be to 1 dp, but not to 2 dp (experimental part).
- 2) Moderate English changes required. There are grammar and orthographical errors in the manuscript, which should be corrected.

My decision is minor revision.

Reviewer: 2

Comments to the Author(s)

The paper of Fu and co-workers deals with the synthesis and cytotoxic activity of glucopyranosyl bearing benzyl derivatives. The main issue with this paper is that the synthesized compounds were tested as mixtures of anomers (in different α , β ratios) and not as pure compounds. Because of this, it is almost impossible to compare the cytotoxic activities of the different derivatives and to conclude about possible structure-activity relationships. I suggest that the authors isolate both the anomers and test them independently. Furthermore, I have found a lot of typographical errors and the manuscript bears the stamp of a non-native English speaker. I suggest the authors ask a Native-English speaker to carefully check their manuscript. I have also other corrections that are written below:

- 1) Title: Change "for" to "against"
- 2) Introduction, page 1: Change "5-fluorouracil" to "5-Fluorouracil"
- 3) Introduction, page 1: Change "to limit" for "limiting"
- 4) Introduction, page 1: Change ".1" to ".[1]"
- 5) Introduction, page 1: You write that "Benzyl and its derivatives are important pharmacophore in many anticancer compounds". So I suggest you add a Figure showing the structures of some of these compounds.
- 6) Introduction, page 2: Change "by the "click chemistry"" to "by "click chemistry" (2 times)
- 7) Introduction, page 2: Change "compling" to "coupling"
- 8) Introduction, page 2: Change "conjugate" to "conjugate"; "conjuating" to "conjugating"; "conjuate" to "conjugate"
- 9) Introduction, page 2: At the end of the introduction section, I would be clearer saying that your target compounds were the one with modification at C2.
- 10) Results and discussion, page 2: "8a-i" must be bolded; "4" must be bolded
- 11) Results and discussion, page 2: Change "itermediate" to "intermediate"; "avaliable" to "available"; "chromatagraphy" to "chromatography"
- 12) Results and discussion, page 2: Change "by the deacetylation" to "by deacetylation"
- 13) Results and discussion, page 2: You write that "All the products were purified by silica gel chromatography". I would add by which methods the compounds were characterized.
- 14) Scheme 1, page 3: For compounds 3 and 4, what was the anomeric ratio? Why was only compound 9d deprotected?

- 15) Scheme 1 caption: Change "Conditions of reactions" to "reaction conditions"; "4" must be bolded. Add a "space" between "4" and "h". Delete the "space" between "N," and "N"
- 16) Title of section 3.2: I would rather write "The cytotoxicity of the synthesized compounds against..."
- 17) Section 3.2, page 3: Change "colorectal" to "colorectal"
- 18) Section 3.2, page 3: Please define "MTS"; Change "via MTS" to "via a MTS". Deleted "the" before "cancer cells"; Change "to HCT-116" to "toward HCT-116"; Change "antiproliferation activity is fairly" to "antiproliferative activity was fairly". Delete "," after "8b-8i"; Change "showed 23 fold" to "showed a 23-fold"
- 19) Table 1: It would have been interesting to test peracetylated glucose as a control. The standard deviations \pm SD must be written with only one significant figures and the values of the IC50 must be rounded up accordingly. For instance, "4.23 \pm 1.98" should be written as "4 \pm 2" and so on for the other values.
- 20) Page 4: Sections 3.3 and 3.4 are identical!!! Please delete one section and change section numbering accordingly.
- 21) Page 4, section 3.3: Change "aganist" to "against"; "As showed" to "As shown"; "Figure 1A, 1C and 1D" to "Figures 1A, 1C, and 1D"; "8d" must be bolded. Change "Figure 1C & 1D" to "Figures 1C and 1D"
- 22) Conclusion: Change "aganist" to "against" (2 times); "selectiity" to "selectivity"; "nomal" to "normal"; "substitution" to "substitution"; "para-position" to "para position"; "influences" to "influenced"; "theapeutic" to "therapeutic"; "is from inducing" to "was shown to induce"
- 23) Section 5.1: Change "propargyl" to "Propargyl"; Change "and removed" by "and solvents were removed".
- 24) Section 5.1: You write that "purities of the intermediates were established by TLC"!!! It is only possible to assess qualitatively the purity by TLC. The proper way to do it is by HPLC and/or elemental analyses.
- 25) Section 5.1: Change "Thin Layer" to "Thin layer"; "water used was distilled" to "water was distilled"
- 26) Section 5.1: Add a "space" between "400" and "MHz" and between "100" and "MHz" (please correct this anywhere it appears in the manuscript"
- 27) Experimental: Please add the aspect of all compounds. The letter "D" in sugars must be written in small capital letters. The "O" must be italicized in compound names. Delete the additional "space" between "of" and "compound"; add an "s" to "compound 7b-7i"
- 28) Experimental: Change "3-bromopropyne" to "3-Bromopropyne"; "completeion" to "completion"; "silca gel" to "silica gel"
- 29) Experimental: Change "compound 8a \square 8i" to "compound 8a-8i"; add a "space" between "1" and "mL" ; delete the "space" between "N," and "N-dimethylformamide"; change "The organic layers were dried" by "The organic layer was dried"; change "sodium chloride" to "saturated aqueous sodium chloride solution"
- 30) Experimental: Delete the unneeded "spaces" in compound names.
- 31) Section 5.6: Change "the incubated compound 8d" to "then compound 8d was incubated"; change "after fixed" to "after being fixed"
- 32) Section 5.7: Change "following treated" to "following treatment"
- 33) Section 5.9: Add a "space" between "5.9" and "Annexin"
- 34) Supporting information: Change "HR-MS" to HRMS"; α , β anomeric ratios must be written for each compound such as for 9b. For 9b, the α , β anomeric ratio is written but this is an equilibrium. Was the NMR spectrum taken at equilibrium?

Author's Response to Decision Letter for (RSOS-201642.R0)

See Appendix A.

Decision letter (RSOS-201642.R1)

This year has been very difficult for everyone, and we want to take the opportunity to thank you for your continued support in 2020.

The Royal Society Open Science editorial office will be closed from the evening of Friday 18 December 2020 until Monday 4 January 2021. We will not be responding during this time. If you have received a deadline within this time period, please contact us as soon as possible to allow us to extend the deadline. If you receive any automated messages during this time asking you to meet a deadline, we offer apologies and invite you to respond after the festive period or during normal working hours.

With our best for a peaceful festive period and New Year, and we look forward to working with you in 2021.

Dear Dr Fu:

Title: Synthesis and Pharmacological Characterization of Glucopyranosyl-conjugated Benzyl Derivatives as Novel Selective Cytotoxic Agents for Colon Cancer
Manuscript ID: RSOS-201642.R1

It is a pleasure to accept your manuscript in its current form for publication in Royal Society Open Science. The chemistry content of Royal Society Open Science is published in collaboration with the Royal Society of Chemistry. I apologise this has taken longer than usual.

On behalf of the Subject Editor Professor Anthony Stace and the Associate Editor Dr Andrew Harned.

RSC Associate Editor
Comments to the Author:

The authors have responded to all points raised by the previous review. I believe this manuscript is now suitable for publication.

Appendix A

November.22, 2020

Dear Dr. Laura Smith,

We are submitting the revised manuscript (RSOS-201642) entitled " Synthesis and Pharmacological Characterization of Glucopyranosyl-conjugated Benzyl Derivatives as Novel Selective Cytotoxic Agents for Colon Cancer ". We thank both reviewers and editors for their constructive comments and suggestions based on which the manuscript has been revised. As suggested by reviewers, we performed some new experiments on the effect of compounds **4** and 2-amino-1,3,4,6-O-acetyl-D-glucose(**AOAG**) as peracetylated glucose derivatives. The results showed that **4** and **AOAG** had almost no cytotoxicity toward HCT-116 and 293T.

We separated all the compounds by preparation thin layers. Two different anomers were isolated. When two anomers were tested by ^1H NMR, respectively, we found that they were still the mixture of α , β anomers again in every anomer sample. The ratio of α/β anomers were fixed. There was an dynamic equilibrium among anomers. The HPLC purities of partial compounds compounds **8a-8i** were attached in supporting information.

We have changed the anomer ratios according to the suggestions of Reviewer 2 provided in the experimental section.

Some minor points suggested by reviewers has been revised. All changes are marked in blue. We also responded point-by-point to all questions raised by reviewers. We hope the revised version will meet the requirement for publication in Royal Society Open Science.

We will enclose our responses with the resubmission. This resubmission has been approved by all authors. We declare that this manuscript has not been published or being published previously in whole or in part elsewhere.

Thank you very much for allowing us to revise the manuscript.

Yours Sincerely,

Boqiao Fu

College of Chemistry and Material Sciences, Hubei Engineering University

Xiaogan 432000, China

E-mail: hubeibridge75@sina.com

Point-by-Point responses to reviewer's comments:

The authors thank both reviewers and editors for their constructive comments and suggestions. The manuscript has been revised accordingly. Our point-by-point response to all comments is given below.

RSC Associate Editor:

Comments to the Author:

The authors have expressed enthusiasm for the work as a whole, but believe the manuscript could be improved by further editing. In addition, Reviewer 2 has

requested that all compounds be tested again as single anomers, rather than the anomeric mixtures that were already tested. It is not clear to me how feasible this will be, or how improved the results would be by doing this. Rather than attempting what would likely be a tedious separation for every compound, I suggest performing this purification/reanalysis of compound 8d (much like the authors already did by testing 9d). I also agree with Reviewer 2 that judging purities by TLC is not sufficient. At the very least, HPLC purities should be reported (I do note that the provided NMR spectra do show the compounds are of relatively high purity).

In addition to the extensive grammatical and typographical corrections suggested by Reviewer 2, I suggest changing the anomer ratios provided in the experimental section so that they all have the same format, e.g. 2.7:1 instead of 100:37. This will make it easier for readers to compare the different ratios.

Reviewers' Comments to Author:

Reviewer: 1

Comments to the Author(s)

The manuscript RSOS-201642 is devoted to the synthesis of novel Glucopyranosyl conjugated benzyl derivatives as potential anticancer agents and can be interesting to the specialists working in this field. The reviewed article is interesting and the theme of the article meets the scope of the journal. Work is performed at sufficient scientific level and has good quality; the results of investigation are professionally interpreted. However, it needs major revision before publication.

To improve the quality and perception of the manuscript I would suggest paying attention to following comments:

My decision is minor revision.

1) All ^{13}C NMR should be to 1 dp, but not to 2 dp (experimental part).

Answer: Thank you for good suggestions. The results of all the ^{13}C NMR have been changed into 1dp in experimental part.

2) Moderate English changes required. There are grammar and orthographical errors in the manuscript, which should be corrected.

Answer: Thank you for good suggestions. English in the manuscript has been revised. Grammar and orthographical errors in the manuscript have been corrected.

Reviewer: 2

Comments to the Author(s)

The paper of Fu and co-workers deals with the synthesis and cytotoxic activity of

glucopyranosyl bearing benzyl derivatives. The main issue with this paper is that the synthesized compounds were tested as mixtures of anomers (in different α,β ratios) and not as pure compounds. Because of this, it is almost impossible to compare the cytotoxic activities of the different derivatives and to conclude about possible structure-activity relationships. I suggest that the authors isolate both the anomers and test them independently. Furthermore, I have found a lot of typographical errors and the manuscript bears the stamp of a non-native English speaker. I suggest the authors ask a Native-English speaker to carefully check their manuscript. I have also other corrections that are written below:

1) Title: Change “for” to “against”

Answer: Thanks the reviewer for the good suggestions, we have changed “for” to “against” in title.

2) Introduction, page 1: Change “5-fluorouracil” to “5-Fluorouracil”

Answer: Thanks the reviewer for the good suggestions, we have changed “5-fluorouracil” to “5-Fluorouracil” in introduction.

3) Introduction, page 1: Change “to limit” for “limiting”

Answer: Thanks the reviewer for the good suggestions, we have changed “to limit” for “limiting” in introduction.

4) Introduction, page 1: Change “.1” to “[1]”

Answer: Thanks the reviewer for the good suggestions, we have changed “.1” to “[1]” in introduction, page 1.

5) Introduction, page 1: You write that “Benzyl and its derivatives are important pharmacophore in many anticancer compounds”. So I suggest you add a Figure showing the structures of some of these compounds.

Answer: Thanks the reviewer for the good suggestions. Figure 1 showing the structures of some anticancer compounds with benzyl as pharmacophore (page 1:) has been inserted into Introduction .

6) Introduction, page 2: Change “by the “click chemistry”” to “by “click chemistry” (2 times)

Answer: Thanks the reviewer for the good suggestions. We have changed “by the “click chemistry” to “by “click chemistry” (2 times) in introduction, page 2.

7) Introduction, page 2: Change “compling” to “coupling”

Answer: Thanks the reviewer for the good suggestions. We have changed “compling” to “coupling” in Introduction, page 2.

8) Introduction, page 2: Change “conjuagate” to “conjugate”; “conjuating” to “conjugating”; “conjuate” to “conjugate”

Answer: Thanks the reviewer for the good suggestions. We have changed “conjuagate” to “conjugate”; “conjuating” to “conjugating”; “conjuate” to “conjugate” in introduction, page 2.

9) Introduction, page 2: At the end of the introduction section, I would be clearer saying that your target compounds were the one with modification at C2.

Answer: Thanks the reviewer for the good suggestions. We have revised that our target compounds were the one with modification at C2 at the end of the introduction section page 2.

10) Results and discussion, page 2: “8a-i” must be bolded; “4” must be bolded

Answer: Thanks the reviewer for the good suggestions. We have bolded “8a-i” and “4” in Results and discussion, page 2.

11) Results and discussion, page 2: Change “itermediate” to “intermediate”; “avaliable” to “available”; “chromatagraphy” to “chromatography”

Answer: Thanks the reviewer for the good suggestions. We have changed “itermediate” to “intermediate”; “avaliable” to “available”; “chromatagraphy” to “chromatography” in Results and discussion, page 2.

12) Results and discussion, page 2: Change “by the deacetylation” to “by deacetylation”

Answer: Thanks the reviewer for the good suggestions. We have changed “by the deacetylation” to “by deacetylation” in Results and discussion, page 2.

13) Results and discussion, page 2: You write that “All the products were purified by silica gel chromatography”. I would add by which methods the compounds were characterized.

Answer: Thanks the reviewer for the good suggestions. We have added all the compounds were charaterized by ^1H NMR, ^{13}C NMR and HRMS in) results and discussion, page 2.

14) Scheme 1, page 3: For compounds 3 and 4, what was the anomeric ratio? Why was only compound 9d deprotected?

Answer: Thanks the reviewer for the good questions. For compounds **3** and **4**, compound **3** was used for the next step directly without further purification. The anomeric ratio of compound **4** was $\beta : \alpha = 3 : 1$.

Why was only compound **9d** deprotected?

Answer: Thanks the reviewer for the good questions. Compound **8d** showed the best biological activities among compounds **8a-8d**. So we chose compound **8d** to be deprotected. The result showed that acetyl group was helpful to increase the antitumor activity via improving liposolubility, which was correspondent with the literature(DOI: 10.1002/jhet.2381).

15) Scheme 1 caption: Change “Conditions of reactions” to “reaction conditions”; “4” must be bolded. Add a “space” between “4” and “h”. Delete the “space” between “N,” and “N”

Answer: Thanks the reviewer for the good suggestions. We have changed “Conditions of reactions” to “reaction conditions”; bolded “4” ; we also have added a “space” between “4” and “h”; and we have deleted the “space” between “N,” and “N” in Scheme 1 caption.

16) Title of section 3.2: I would rather write “The cytotoxicity of the synthesized compounds against...”

Answer: Thanks the reviewer for the good suggestions. We have changed “The cytotoxicity of the synthesized compounds against HCT-116 and 293T” in title of section 3.2:

17) Section 3.2, page 3: Change “coloreactal” to “colorectal”

Answer: Thanks the reviewer for the good suggestions. We have changed “coloreactal” to “colorectal” in section 3.2.

18) Section 3.2, page 3: Please define “MTS”; Change “via MTS” to “via a MTS”. Deleted “the” before “cancer cells”; Change “to HCT-116” to “toward HCT-116”; Change “antiproliferation activity is fairly” to “antiproliferative activity was fairly”. Delete “,” after “8b-8i”; Change “showed 23 fold” to “showed a 23-fold”

Answer: Thanks the reviewer for the good suggestions. MTS is working solution containing 3-(4,5-dimethylthiazol-2-yl)-5-(3-carboxymethoxyphenyl)-2-(4-sulfo

phenyl) -2H-tetrazolium (MTS, 2 mg/mL) and phenazine methosulfate (PMS, 0.92 mg/mL) that were diluted in Dulbecco's phosphate-buffered saline (DPBS)(MTS: Promega, Catalog No. G1111; MTS: Promega, Catalog No. G1111; PMS: Sigma, Catalog No. P9625, for example, DOI: 10.3892/ol.2020.12175); We have changed “via MTS” to “via a MTS” and deleted “the” before “cancer cells”. We have changed “to HCT-116” to “toward HCT-116” and changed “antiproliferation activity is fairly” to “antiproliferative activity was fairly”. in Section 3.2, page 3.

19) Table 1: It would have been interesting to test peracetylated glucose as a control. The standard deviations \pm SD must be written with only one significant figures and the values of the IC₅₀ must be rounded up accordingly. For instance, “4.23 \pm 1.98” should be written as “4 \pm 2” and so on for the other values.

Answer: Thanks the reviewer for the good suggestions. We have tested peracetylated glucose **4** and 2-amino-1,3,4,6-O-acetyl-D-glucose(**AOAG**) as a control.

About the suggestion: The standard deviations \pm SD must be written with only one significant figures and the values of the IC₅₀ must be rounded up accordingly.

Answer: Thanks the reviewer for the good suggestions. Actually in order to make a good recognition in the data, the standard deviations \pm SD and the values of the IC₅₀ were written with at least two significant figures. For example the values of IC₅₀ in the papers in Royal Society Open Science (DOI:10.1098/rsos.171510 and 10.1098/rsos.200050).

20) Page 4: Sections 3.3 and 3.4 are identical!!! Please delete one section and change section numbering accordingly.

Answer: Thanks the reviewer for the good suggestions. We have deleted section 3.4 and changed section numbering accordingly.

21) Page 4, section 3.3: Change “aganist” to “against”; “As showed” to “As shown”; “Figure 1A, 1C and 1D” to “Figures 1A, 1C, and 1D”; “8d” must be bolded. Change “Figure 1C &1D” to “Figures 1C and 1D”

Answer: Thanks the reviewer for the good suggestions. We have changed “aganist” to “against”; “As showed” to “As shown”; “Figure 1A, 1C and 1D” to “Figures 1A, 1C, and 1D”; bolded “8d”; and changed “Figure 1C &1D” to “Figures 1C and 1D” in Page 4, section 3.

22) Conclusion: Change “aganist” to “against” (2 times); “selectiity” to “selectivity”; “nomal” to “normal”; “subsitution” to “substitution”; “para-position” to “para position”; “influences” to “influenced”; “theapetuic” to “therapeutic”; “is from inducing” to “was shown to induce”

Answer: Thanks the reviewer for the good suggestions. We have changed “aganist” to “against” (2 times); “selectiity” to “selectivity”; “nomal” to “normal”; “subsitution” to “substitution”; “para-position” to “para position”; “influences” to “influenced”; “theapetuic” to “therapeutic”; “is from inducing” to “was shown to induce” in Conclusion.

23) Section 5.1: Change “propargyl” to “Propargyl”; Change “and removed” by “and solvents were removed”.

Answer: Thanks the reviewer for the good suggestions. We have changed “propargyl” to “Propargyl”; we have changed “and removed” by “and solvents were removed” in Section 5.1.

24) Section 5.1: You write that “purities of the intermediates were established by TLC”!!! It is only possible to assess qualitatively the purity by TLC. The proper way to do it is by HPLC and/or elemental analyses.

Answer: Thanks the reviewer for the good suggestions. We have changed “purities of the intermediates were established by TLC” to “purities of the intermediates were established by HPLC.” in Section 5.1. We have afforded the HPLC results of target compounds in supporting information.

25) Section 5.1: Change “Thin Layer” to “Thin layer”; “water used was distilled” to “water was distilled”

Answer: Thanks the reviewer for the good suggestions. We have changed “Thin Layer” to “Thin layer”; “water used was distilled” to “water was distilled” in Section 5.1.

26) Section 5.1: Add a “space” between “400” and “MHz” and between “100” and “MHz” (please correct this anywhere it appears in the manuscript”

Answer: Thanks the reviewer for the good suggestions. We have added a “space” between “400” and “MHz” and between “100” and “MHz” in Section 5.1 and also corrected this anywhere it appears in the manuscript”.

27) Experimental: Please add the aspect of all compounds. The letter “D” in sugars must be written in small capital letters. The “O” must be italicized in compound names. Delete the additional “space” between “of” and “compound”; add an “s” to “compound 7b-7i”

Answer: Thanks the reviewer for the good suggestions. We have added the aspect of all compounds. The letter “D” in sugars has been written in small capital letters. The

“O” has been italicized in compound names. We have deleted the additional “space” between “of” and “compound”; we also have added an “s” to “compound 7b-7i” in Experimental Section.

28) Experimental: Change “3-bromopropyne” to “3-Bromopropyne”; “completeion” to “completion”; “silca gel” to “silica gel”

Answer: Thanks the reviewer for the good suggestions. We have changed “3-bromopropyne” to “3-Bromopropyne”; “completeion” to “completion”; “silca gel” to “silica gel” in Experimental Section.

29) Experimental: Change “compound 8a ~8i” to “compound 8a-8i”; add a “space” between “1” and “mL”; delete the “space” between “N,” and “N-dimethylformamide”; change “The organic layers were dried” by “The organic layer was dried”; change “sodium chloride” to “saturated aqueous sodium chloride solution”

Answer: Thanks the reviewer for the good suggestions. We have changed “compound 8a ~8i” to “compound 8a-8i”; added a “space” between “1” and “mL”; and deleted the “space” between “N,” and “N-dimethylformamide”; changed “The organic layers were dried” by “The organic layer was dried”; changed “sodium chloride” to “saturated aqueous sodium chloride solution” in Experimental Section.

30) Experimental: Delete the unneeded “spaces” in compound names.

Answer: Thanks the reviewer for the good suggestions. We have deleted the unneeded “spaces” in compound names in Experimental Section.

31) Section 5.6: Change “the incubated compound 8d” to “then compound 8d was incubated”; change “after fixed” to “after being fixed”

Answer: Thanks the reviewer for the good suggestions. We have changed “the incubated compound 8d” to “then compound 8d was incubated” and changed “after fixed” to “after being fixed” in Section 5.6.

32) Section 5.7: Change “following treated” to “following treatment”

Answer: Thanks the reviewer for the good suggestions. We have changed “following treated” to “following treatment” in Section 5.7.

33) Section 5.9: Add a “space” between “5.9” and “Annexin”

Answer: Thanks the reviewer for the good suggestions. We have added a “space” between “5.9” and “Annexin” in Section 5.9.

34) Supporting information: Change “HR-MS” to HRMS”; α,β anomeric ratios must be written for each compound such as for 9b. For 9b, the α,β anomeric ratio is written but this is an equilibrium. Was the NMR spectrum taken at equilibrium?

Answer: Thanks the reviewer for the good suggestions. We have changed “HR-MS” to “HRMS”; α,β anomeric ratios have been written for each compound such as for 9b.

For 9b, the α,β anomeric ratio is written but this is an equilibrium. Was the NMR spectrum taken at equilibrium?

Answer: Thanks the reviewer for the good questions. The NMR spectrum was taken at equilibrium. Even we separated every compound α,β anomeric by preparation thin layers with the eluant(PE/EA =1:1 -1:2 for compounds **8b-8d**, PE/EA =1:4 for compound **8a**; CH₂Cl₂: MeOH = 10: 1 for compound **9d**). When α anomer or β anomer was tested by NMR, respectively, we found the results surprized us that every sample was still the mixture of α/β anomers, and the ratio was fixed. It supported that we believed that there was an equilibrium in the compounds. These results were due to the absence of neighboring group participation to lock the configuration.